# A LAMP Detection System Based on a Microfluidic Chip for *Pyricularia grisea*

**DOI:** 10.3390/s25082511

**Published:** 2025-04-16

**Authors:** Chenda Wu, Jianing Cheng, Yinchao Zhang, Ping Yao

**Affiliations:** 1College of Information and Electrical Engineering, Shenyang Agricultural University, Shenyang 110866, China; chendawu@stu.syau.edu.cn (C.W.); yinchaozhang@stu.syau.edu.cn (Y.Z.); 2College of Bioscience and Biotechnology, Shenyang Agricultural University, Shenyang 110866, China; 2022220017@stu.syau.edu.cn; 3Liaoning Engineering Research Center for Information Technology in Agriculture, Shenyang 110866, China

**Keywords:** microfluidic chip, LAMP, micro-mixing channels, isothermal control, *Pyricularia grisea*

## Abstract

As one of the major rice fungal diseases, blast poses a serious threat to the yield and quality of rice globally. It is caused by the pathogen *Pyricularia grisea*. Therefore, the development of rapid, accurate, and portable microfluidic detection system for *Pyricularia grisea* is important for the control of rice blast. This study presents an integrated microfluidic detection system for the rapid and sensitive detection of *Pyricularia grisea* using the LAMP detection method. The microfluidic detection system includes a microfluidic chip, a temperature control module, and an OpenMv camera. The micro-mixing channels with shear structures improve the mixing efficiency to about 98%. Flow-blocking valves are used to reduce reagent loss in the reaction chamber. The temperature control module is used to heat the reaction chamber, maintaining a stable temperature of 65 °C. The microfluidic chip detection chamber is used for image inspection using an OpenMv camera. The developed system can detect *Pyricularia grisea* in the range of 10 copies/μL–10^5^ copies/μL within 45 min. Specificity and interference experiments were performed on *Pyricularia grisea*, validating the method’s good reliability. This LAMP detection system based on a microfluidic chip has strong potential in the early and effective detection of rice blast.

## 1. Introduction

Rice is one of the world’s important food crops, especially in developing countries [1,2], providing the staple food for more than four billion people [3]. Rice blast, commonly known as “rice cancer” [4], is a worldwide fungal hazard and the most important fungal disease in rice cultivation [5,6]. The pathogen is *Pyricularia grise*, which is highly contagious, posing a serious threat to the yield and quality of rice [7,8,9]. *Pyricularia grise* mainly overwinters in the form of conidia and mycelium by attaching itself to straw and rice grains. In the following year, conidia proliferates and spreads to adjacent crop tissues through wind and rain, eventually resulting in the formation of diseased plants [10]. Conidia that is attached to the diseased plants can infest again with the help of wind and rain, ultimately leading to widespread rice blast during rice cultivation. Therefore, the early detection of *Pyricularia grisea* is crucial for controlling the spread of rice blast disease. The current detection methods for *Pyricularia grise* are biomolecular testing [11,12], spectral detection [6,13], and manual identification under a microscope [14]. Although spectral detection has high accuracy, training the model requires a long time and many samples. Meanwhile, optimizing the algorithm model demands strong professional expertise. The method of manual identification under the microscope is time consuming and labor intensive, and it may also lead to counting errors. Biomolecular testing is categorized into protein molecular-level testing and gene-level testing. Protein molecular-level testing mainly involves enzyme-linked immunosorbent assays (ELISA), which can produce false-positive results [15]. At the genetic level, nucleic acid molecular techniques such as polymerase chain reaction (PCR) and loop-mediated isothermal amplification (LAMP) are widely used for pathogen detection. Both PCR and LAMP provide accurate detection via target DNA amplification, but PCR requires a complex thermal cycling process for its amplification. As an isothermal amplification technique, LAMP offers advantages such as high detection sensitivity, the elimination of thermal cycling, and high specificity. By introducing primers, polymerases, dye, and reagent substrates, LAMP technology can detect target DNA within 1 h at 62–65 °C. Chen et al. [16] developed a digital LAMP platform to quantify Vibrio parahaemolyticus nucleic acids by enumerating fluorescence spots on a porous membrane. The proposed system can detect 100 copies of target DNA within 30 min at 65 °C. Li et al. [17] established a LAMP detection system by designing an amplification primer, which could detect the genetic mutants in rice in 60 min at a 62 °C. The detection limit is 100 fg/μL, which is 10 times higher than that of conventional PCR. LAMP can be an effective technique for detecting *Pyricularia grisea*.

A microfluidic chip is a powerful tool used for sensitive, high-speed, and low-cost analysis [18], which has the advantages of small size, low reagent consumption, operational automation, and rapid detection capabilities [19,20]. Currently, many LAMP analyses based on various microfluidic chips are used for the accurate and rapid detection of pathogens [21]. Wan et al. [22] developed a digital microfluidic device for the LAMP DNA detection of Trypanosoma brucei in a micro-supply sample. Yao et al. [23] proposed a fast-loading microfluidic chip for vector-borne viruses detection using RT-LAMP, and the detection limits of all eight viruses were less than 500 copies. It is possible to combine microfluidic chips with LAMP technology to achieve the accurate detection of *Pyricularia grisea*.

In this study, an integrated biosensing system based on a microfluidic chip is designed for the quantitative detection of *Pyricularia grisea* using LAMP, as shown in Figure 1. The biosensing system includes a microfluidic chip, a temperature control module, and an OpenMv detection module. The homogeneous mixing of reagents is critical for enhancing amplification efficiency [24]. Micro-mixing channels were designed, optimized, and fabricated for mixing LAMP reagents and *Pyricularia grisea* DNA samples. The microfluidic chip reaction chamber is used for the amplification of reagents, which incorporates flow-blocking valves to reduce reagent loss. The ceramic heater in the temperature control module is designed to heat the reaction chamber at 65 °C for isothermal amplification. The microfluidic chip detection chamber can be imaged using the OpenMv camera. A stepper motor is used to regulate the vertical gearing to move the control valve up and down. The control valve is placed above the flow-blocking valves of the microfluidic chip. To avoid the influence of variable light, a black box is fabricated using a 3D printer. The microfluidic chip is placed inside the black box for amplification and detection. The black box contains a light bead and a flat mirror, which make the detection chamber of the chip brighter. The images of the detection chamber are captured and transferred to a computer. OpenMv IDE 4.2.0 software is used to analyze R, G, and B data. The proposed microfluidic detection system can detect *Pyricularia grisea* with a minimum of 10 copies/μL within 45 min.

The process of detecting *Pyricularia grisea* using the microfluidic detection system is as follows. First, the temperature control module is activated 3 min in advance to preheat the chip’s reaction chamber to 65 °C. Then, as shown in Figure 2b, the *Pyricularia grisea* DNA solution and LAMP reagents are added into the chip’s reservoirs. Next, the chip’s inlets are connected to the micro-injector’s tubing as shown in Figure 2a. After that, the micro-injector is set to 0.4 μL/s to inject the reagents into the micro-mixing channels. The regents enter the reaction chamber after passing through the channels, with a total mixing time of 4 min as shown in Figure 2c. Then, the flow blocking device is activated, as shown in Figure 2e. The linkage bar moves downward, causing the switch to close the flow blocking microwaves. This cuts off the micro channels on both sides of the reaction chamber, sealing the reagents inside. After 45 min of amplification, as shown in Figure 2d, the blocking device is removed to open the valves and allow the reagents to flow into the detection chamber. Finally, the OpenMv IDE software is opened and the OpenMv camera shown in Figure 2f is used to capture images of the detection chamber, outputting real-time R, G, and B values to the computer. The RGB information of the images can reflect the DNA concentrations of *Pyricularia grisea*.

## 2. Materials and Methods

### 2.1. LAMP Detection Principle

The LAMP reaction detection principle for *Pyricularia grisea* DNA is shown in Figure 3. During the LAMP amplification reaction, genomic DNA of *Pyricularia grisea* undergoes strand displacement amplification under isothermal conditions. Through the coordinated activity of specifically designed primers, reaction buffers, and Bst DNA polymerase, the DNA double strands unwind during heating, forming dumbbell-shaped DNA structures. The dumbbell-shaped DNA structures have multiple start sites, and the amplification of the DNA is initiated at these multiple start sites. The products of amplification can be accumulated rapidly within 40 min. The volume of this LAMP reaction system is 20 μL, which contains 14 μL of the template DNA solution, 5 μL of reaction buffers, and 1 μL of primers. The reaction buffers contain Skygen LAMP dye markers that react with the amplification products. When the dye markers react with the amplification products, the dye markers turn sky blue.

### 2.2. Reagents and Instruments

*Pyricularia grisea* acts as the target pathogen, and other species (*Rhizoctonia solani*, *Xanthomonas oryzae*, *Bipolaris oryzae*, and *Ustilaginoidea virens*) are the non-target pathogens in the experimental setup. All the rice pathogens were obtained from the School of Plant Protection, Shenyang Agricultural University. The LAMP amplification kit (LAMP Master Mix, Bst DNA polymerase 2.0, LAMP primer dry powder, ultrapure water, and LAMP purified DNA solution) and electrophoresis solution were provided by Beijing Tianjingsha Gene Technology. Naphthol Green B was purchased from Tianjin Zhiyuan Chemical Reagent Co. (Tianjin, China). Amplification experiments were performed using a Q1600 real-time fluorescence quantitative PCR instrument (Hangzhou Boheng Technology Co., Ltd., Hangzhou, China). A Formlab 3D printer (Beijing Aike Hengtong Science and Technology Development Co., Ltd., Beijing, China) was used for mold making. The materials for the microfluidic chip manufacture are a transparent glass plate, double-sided adhesive, and PDMS (Dow Corning, Midland, MI, USA). A Bambu Lab P1S 3D printer (Bambu Lab Shenzhen Top Bamboo Technology Co., Ltd., Shenzhen, China) was used for black box fabrication. The ceramic heater, the step motor, and the OpenMv camera were purchased from Guangzhou Star Pupil Information Technology Co., Ltd., Guangzhou, China.

The specific LAMP amplification reaction to *Pyricularia grisea* DNA relies on specific primers. The LAMP primers were designed via the LAMP Primer Design Website (https://lamp.neb.com), using information from 1 to 2000 genes in chromosome 6 of the *Pyricularia grisea* T3 strain. Table 1 shows the primer information of the screened *Pyricularia grisea*.

### 2.3. Micro-Mixing Channels Modeling

When performing LAMP amplification reactions on a microfluidic chip, the complete mixing of the reagents can significantly enhance amplification efficiency. The models of the micro-mixing channels were analyzed and optimized using COMSOL Multiphysics 6.1 finite element simulation software. Models were established based on diffusion and hydrodynamic theories, as shown in Figure 4. The triaging, extending, disturbing, and shearing effect in the micro-mixing channels accelerate the fluid diffusion process, giving the micro-mixing channels good mixing efficiency. As shown in Figure 4a, the four square micro-mixing channels not only continuously perturb the fluid but also extend its flow distance, thereby increasing the diffusion rate and the contact time between fluid constituents, which enhances the mixing efficiency. As shown in Figure 4b, a micro-channel model was developed by substituting a square channel with dual rectangular buffer channels in the original square channel structures. The transition of fluid flows from the square channels to the buffer channels enhances perturbation and increases the interfacial contact area among fluid constituents, augmenting the mixing efficiency. As shown in Figure 4c, the design of Figure 4b was optimized by integrating three shear structures within the channels. Through the shear structures, fluid constituents are sheared, stretched, and deformed to enhance the mixing efficiency.

Given that the total volume of the reaction reagents is 20 μL, the width and height of the micro channels are typically designed to not exceed 1 mm. Due to the extremely low Reynolds number of fluids within the micro-mixing channels, the fluid is in a laminar flow state. Using the above-designed micro-mixing channels models, the mixing efficiency of each model was simulated in the flow rate range from 0.1 μL/s to 1 μL/s by using simulation software. The results were used to guide the subsequent modeling and chip fabrication.

### 2.4. Microfluidic Chip Design and Fabrication

The microfluidic chip consists of inlets, reservoirs, mixing channels, a reaction chamber, flow-blocking valves, a detection chamber, and an outlet. Figure 5 shows the designed structures and the fabricated chip. The microfluidic chip is fabricated by bonding the PDMS layer to the glass substrate with double-sided adhesive. The larger reservoir is used to load the DNA template solution, while the smaller reservoir is used to load the LAMP reaction reagents. After the reagents are fully mixed in the mixing channels, they flow into the reaction chamber. The flow-blocking valve is designed to seal the reagents inside the reaction chamber. After the reaction is complete, the valves are turned on. The reagents flow into the detection chamber, where images are captured by the OpenMv camera. Figure 5a,b shows the structures and dimensions of the microfluidic chip. The dimensions of the PDMS layer are 80 mm length × 40 mm width × 4 mm height, and the depth of the channels and chambers is 1 mm. Figure 5c shows an image of the microfluidic chip.

The microfluidic chip was fabricated using the PDMS molding method. SolidWorks 2022 software was used to design the 3D structure of the chip. The designed microfluidic chip mold was fabricated using a Formlabs 3D printer. Silicone elastomer PDMS agent and the curing agent were thoroughly mixed in a 10:1 mass ratio, and then the mixed PDMS was poured into the mold. The mold with mixed PDMS was placed into a vacuum oven. The vacuum pump is turned on to completely extract the air from the PDMS. Then, the temperature of the oven is set to a constant 45 °C. After 12 h, the PDMS was fully cured. After demolding the cured PDMS, the PDMS layer was bonded to the glass substrate with double-sided transparent adhesive. After the microfluidic chip was fabricated, anhydrous ethanol was used to modify the microchannels and chambers, preventing non-specific adsorption.

### 2.5. Hardware Design

The hardware design includes a flow-blocking device, a temperature control module, and a detection module. When the reagents are injected into the reaction chamber, the flow-blocking device prevents the reagents from flowing out of the reaction chamber. The temperature control module maintains the temperature in the reaction chamber at 65 °C. The detection module captures the images of the detection chamber by the OpenMv camera.

#### 2.5.1. Flow-Blocking Device

A screw-driven stepper-motor-based flow-blocking device was proposed. It works with micro-valves in the microfluidic chip to seal the reaction reagents in the chip’s reaction chamber. The device’s linkage bar, micro-valve switch, and motor fixture were designed using SolidWorks. Figure 6 shows the models of these components and the assembled device. The linkage bar, micro valves switch, and screw motor are screwed together, with the motor installed in the motor fixture. When the motor runs, it drives the slider vertically, moving the linkage bar and micro-valve switch to open or close the micro-valves.

After the reagents flow into the reaction chamber, a flow-blocking device is used to seal the reagents within the reaction chamber. The device is controlled by a stepper motor, and the moving accuracy of the vertical gearing is 0.1 mm. The collaborative operation of the flow-blocking device and the microfluidic chip is shown in Figure 7. When the reagents are inside the reaction chamber, the motor is activated to press downward on the two micro valves to cut off the channels. After the amplification reaction is completed, the motor is activated to remove the control valve upward, and the reagents flow into the detection chamber.

#### 2.5.2. Temperature Control Module

The temperature control module contains a microcontroller, a circular ceramic heater (R = 9 mm) and a miniature temperature sensor. In order that the temperature is evenly distributed in the reaction chamber, the ceramic heater is placed underneath the reaction chamber. The STM32F103 microcontroller (Guangzhou Xingyi Electronics Technology Co., Ltd., Guangzhou, China), integrated with a fuzzy PID control algorithm, is used to keep the temperature steady.

There is a glass substrate between the ceramic heater and the reaction chamber, so the temperature of the ceramic heater is different from that in the reaction chamber. It is necessary to calibrate the temperature of the ceramic heater to maintain the temperature of the reaction chamber at 65 °C. Temperature calibration experiments were designed and performed. According to the experimental results, when the temperature of the ceramic heater is 80.1 °C, the temperature in the reaction chamber can be steadily maintained at 65 °C. The temperature of the reaction chamber can reach 65 °C within 5 min, which shows the temperature control module has a good heating rate. Before the start of the *Pyricularia grisea* detection, the reaction chamber should be pre-heated for 5 min to maintain steady temperature.

#### 2.5.3. Detection Module

The detection module consists of a light bead and an OpenMv camera. The white light bead is 5 mm in diameter. The light bead and flat mirror are used in conjunction inside the black box to make the detection chamber brighter. The OpenMv camera is connected to a computer. The camera can capture images from the detection chamber and send them to the computer in real time. Using OpenMV IDE v 4.2 software, the R, G, and B values of the transmitted images are analyzed by a computer. According to the LAMP experimental principle, the solution appears sky blue. Then B/G and B/R values are calculated and used for curve fitting, which is performed against the DNA concentrations of *Pyricularia grisea*. Between the two fitted curves, the one with better correlation was selected as the standard curve.

## 3. Results and Discussion

### 3.1. LAMP Amplification Efficiency and Specificity

Amplification efficiency and specificity experiments for the LAMP assay were designed. Samples of *Pyricularia grisea* DNA solutions at a concentration of 10^5^ copies/μL were set up as positive controls. Samples of *Rhizoctonia solani* DNA with the same concentration of 10^5^ copies/μL was set as a negative control group. A 20 μL reaction mixture was prepared in PCR tubes, comprising 14 μL of DNA template solution and 6 μL of LAMP reaction reagents. The LAMP reaction reagents consist of 1 μL LAMP amplification primer and 5 μL reaction buffers. The prepared PCR tubes were put into the PCR machine, and the PCR machine was set at a temperature of 65 °C for 60 min. As shown in Figure 8a, the amplification curves were obtained. The amplification process starts at about 15 min, and the four positive samples produce obvious amplification curves. No amplification reactions occurred in the four negative samples. The amplification reaction basically ends when the amplification reaches about 40 min. The reaction time can be optimized to 40 min in the subsequent design of the detection platform. After the completion of the amplification reaction, the amplification products were added to 2% agarose gel electrophoresis. The electrophoresis results are shown in Figure 8b. Odd-numbered channels represent the electrophoretic results of four negative samples, which produce no bright bands. Even-numbered channels represent the electrophoretic results of the four positive samples, which produce bright bands. The bright bands show that the positive samples have been amplified. The amplification curves and electrophoresis results showed that the LAMP method has good amplification efficiency and specificity for *Pyricularia grisea*.

### 3.2. Simulation Analysis of the Micro-Mixing Channels

The three models of micro-mixing channels shown in Figure 4 were simulated using COMSOL Multiphysics 6.1 software. The simulation results are shown in Figure 9a. The horizontal length of the mixing channel is 35 mm in the three models. The horizontal length of each square-shaped channel is 6 mm, and the width of the microchannel is 0.5 mm. The microchannel model with shear structures has the best mixing efficiency, which is used for mold and chip fabrication.

As shown in Figure 9b, the mixing efficiency of each model was simulated at different flow rates ranging from 0.1 μL/s to 1 μL/s. The flow rates of the model are negatively correlated with the mixing efficiency. At the flow rate of 0.1 μL/s, the mixing efficiency of the model simulation with shear structures can reach 99%. However, it should be noted that the solution flow rate is slow at this time and the total mixing time is long. By calculation, the total time of the whole mixing process was 15 min when the fluid inflow rate was 0.1 μL/s. When the fluid inflow rate was 0.4 μL/s, the mixing time was shortened to 3.7 min. After comprehensively considering the mixing efficiency and mixing time, the flow rate was set to 0.4 μL/s. The mixing efficiency was approximately 98%. This mixing efficiency is sufficient to meet the requirements for reagent mixing.

### 3.3. Mixing Experiments

The mixing experiments were performed using naphthol green B reagent and purified water on the mixing chip with shear structures, as shown in Figure 10. We prepared naphthol green B solutions with a concentration of 2.5 g/L as the original solution. Mixing experiments were carried out using 10 μL of purified water and 10 μL of the original Naphthol Green B solution. Figure 10a shows the initial state of the solution on the mixing chip at the start of the mixing experiments. A micro injection pump was used to push 10 μL of purified water and 10 μL of the original Naphthol Green B solution into the micro-mixing channels at the flow rate of 0.4 μL/s. The solution flows into the analyzing area after mixing is complete. As shown in Figure 10c, the average gray value $G_{j}$ of the analyzed area was obtained using ImageJ 1.54p software to calculate the gray level of the image, with a pixel size of 30 × 30 on the analyzed area. We added 2.5 g/L of naphthol Green B original solution to the analysis area of the mixed chip, using the grayscale value of the analysis area at this point as a control value $G_{0}$. We then calculated the mixing efficiency of the micro-mixing channels. The equation for mixing efficiency is as follows:(1)$$E_{j}=\left(1-\left|1-\frac{G_{0}}{G_{j}}\right|\right)\times 100\%$$

The average mixing efficiency of the five experiments was 97.32%. This proves that the actual mixing efficiency of the micro-mixing channels meets the expectations.

### 3.4. Detection System Sensitivity and Standard Curve

*Pyricularia grisea* was detected using the developed system, the range and limit of detection were determined, and standard curves were developed. A DNA solution of *Pyricularia grisea* with a concentration of 10^5^ copies/μL was used to prepare six concentration gradients of DNA samples through a 10-fold serial dilution method. The concentration gradients are 10^5^ copies/μL, 10^4^ copies/μL, 10^3^ copies/μL, 10^2^ copies/μL, 10 copies/μL, and 1 copy/μL. Purified water was used as a negative control. As shown in Figure 11a, the depth of the blue color in the detection chamber of the chip is positively correlated with the concentration grade of the DNA solution. An OpenMv camera was utilized to capture images of the solution in the detection chamber and transfer them to a computer. Because of the depth variation in the blue color presented in the detection chamber, the ratio of its blue channel B to the green channel G and the red channel R (B/G, B/R) and the concentration of the DNA solution show a correlation. Curve fitting of the data against DNA sample concentrations was carried out using Origin 2022 9.9 software. As shown in Figure 11b,c, the correlation coefficient between the fitted curves for B/R and DNA concentration was 0.88, whereas the correlation coefficient between the fitted curves for B/G and DNA concentration was 0.97. Therefore, the fitting curve of B/G to DNA concentration was selected as the standard curve of the detection system. The sensitivity of the detection system for *Pyricularia grisea* is 10 copies/μL, and the detection range is from 10 copies/μL to 10^5^ copies/μL. The standard curve is:(2)$$B/G=0.0675lgC_{P}+0.8842$$

### 3.5. Detection System Reliability Performance Evaluation

Specificity experiments, anti-interference experiments, and cross-reactivity detection accuracy experiments were performed to evaluate the reliability of the detection system.

#### 3.5.1. Detection System Specificity

Specificity experiments were performed on *Pyricularia grisea* using the developed system. *Pyricularia grisea* DNA at a concentration of 10^4^ copies/μL was used as a positive template. *Rhizoctonia solani*, *Xanthomonas oryzae*, *Bipolaris oryazae*, and *Ustilaginoidea virens* at a concentration of 10^4^ copies/μL were used as negative templates. Purified water in the same volume as the positive template was used as a negative control. The results are shown in Figure 12a; only *Pyricularia grisea* had a positive reaction, with a blue color in the chamber.

As shown in Figure 12b, the detection value of *Pyricularia grisea* was significantly higher than that of the negative template and the negative control, which indicates a significant amplification reaction. The detection values of the negative template and the negative control were close to each other, so it was determined that no amplification reaction occurred. The specificity of this detection system is good.

#### 3.5.2. Detection System Anti-Interference

The experiment was designed for validating the anti-interference of the detection system. Equal volumes of the two samples were prepared. The first samples were a *Pyricularia-grisea*-only DNA solution with DNA concentrations of 10^5^ copies/μL, 10^4^ copies/μL, and 10^2^ copies/μL. The second samples were a 50/50 mixture of *Pyricularia grisea* and *Rhizoctonia solani* DNA solutions. The DNA concentrations of *Pyricularia grisea* were 10^5^ copies/μL, 10^4^ copies/μL, and 10^3^ copies/μL. We used purified water as a negative control group. The developed system was used to detect the above samples. As shown in Table 2, the standard deviations of the detection values were 1.85%, 1.95%, and 1.7% for the samples containing only *Pyricularia grisea* and the mixed samples. This standard deviation is relatively small and within the acceptable range. The detection system has relatively good anti-interference capabilities.

#### 3.5.3. Detection Capability of the Detection System Under Cross-Reaction Conditions

Under the conditions of cross-reactivity with four types of negative samples, detection experiments were performed on *Pyricularia grisea* in mixed solutions to verify the system’s stability in such environments. Cross-reaction LAMP systems were formulated using four negative samples: *Rhizoctonia solani*, *Xanthomonas oryzae*, *Bipolaris oryzae*, and *Ustilaginoidea virens*. The preparation process for the mixed solution is as follows:(1)Prepare three PCR tubes, each containing 25 μL of the template dilution solution.(2)To the first tube, add 5 μL each of *Pyricularia grisea*, *Rhizoctonia solani*, *Xanthomonas oryzae*, *Bipolaris oryzae*, and *Ustilaginoidea virens* (10⁶ copies/μL). Mix well for 1 min to obtain a 50 μL solution with 10^5^ copies/μL of *Pyricularia grisea* DNA. Store it in a refrigerated container for later use.(3)To the second tube, add 5 μL of the 10^5^ copies/μL *Pyricularia grisea* mix (from step 2) and 20 μL of the template dilution solution. Mix for 1 min to obtain a 50 μL solution with 10^4^ copies/μL of *Pyricularia grisea* DNA. Store and refrigerate.(4)To the third tube, add 5 μL of the 10^4^ copies/μL Pyricularia grisea mix (from step 3) and 20 μL of the template dilution solution. Mix for 1 min to obtain a 50 μL solution with 10^3^ copies/μL of *Pyricularia grisea* DNA. Store and refrigerate.

Three mixed solutions with *Pyricularia grisea* DNA concentrations of 10^3^ copies/μL, 10^4^ copies/μL, and 10^5^ copies/μL were prepared, each with a volume of 50 μL. The microfluidic detection system was used to detect the mixed DNA solutions for concentration, with each solution detected three times. Quantitative standard curves were used to calculate the concentration values. The statistical results of the detected concentrations are shown in Table 3. Here, C_R_ is the actual concentration of *Pyricularia grisea* DNA in the solution. T_1_, T_2_, and T_3_ are the detected concentration values from the three experiments. A_1_, A_2_, and A_3_ represent the relative accuracy of each experimental detection.

The microfluidic detection system detected mixed DNA solutions of *Pyricularia grisea* at concentrations of 10^3^, 10^4^, and 10^5^ copies/μL; the average relative detection accuracy was 83.43%, 87.69%, and 87.77%, respectively. These results show that the system has good detection stability in cross-reaction conditions.

## 4. Conclusions

In this study, an integrated microfluidic biosensing system was developed for the rapid and accurate detection of *Pyricularia grisea*. The system consists of a microfluidic chip with flow-blocking valves, a temperature control module, and a detection system equipped with an OpenMv camera module. The LAMP method is used for the rapid detection of *Pyricularia grisea*. Modeling and simulation analysis were performed using COMSOL Multiphysics 6.1 software, along with mixing experiments, ultimately resulting in the fabrication of a microfluidic chip with a mixing efficiency of approximately 98%. This chip incorporates all functions for LAMP amplification of *Pyricularia grisea*. The inflow velocity of the reagents was set to 4 × 10^−4^ m/s. A flow-blocking device with 0.1 mm vertical positioning accuracy was designed and fabricated. When the flow-blocking device is activated, it prevents the reagents from flowing out of the reaction chamber. The temperature control module integrated with a fuzzy-PID control algorithm was developed to heat the reaction chamber of the microfluidic chip, maintaining the temperature at 65 °C. Considering the temperature loss of the chip glass substrate, a temperature calibration experiment was set up to correct the temperature of the ceramic heater to 80.1 °C The microfluidic biosensing system shows a detection range of 10 copies/μL to 10^5^ copies/μL for *Pyricularia grisea* at 65 °C within 45 min. Finally, specificity and interference resistance experiments were performed. The specificity experiment was performed on *Pyricularia grisea*, *Rhizoctonia solani*, *Xanthomonas oryzae*, *Bipolaris oryzae*, and *Ustilaginoidea virens.* During the specificity experiment for *Pyricularia grisea*, only the positive sample exhibited amplification, and the system demonstrated good specificity. In the anti-interference experiment, the standard deviations of the detection values were 1.85%, 1.95%, and 1.7%. These standard deviations are within the acceptable range, and the detection system possesses considerable anti-interference capabilities. In the cross-reaction experiments, the microfluidic detection system detected mixed DNA solutions of *Pyricularia grisea* at 10^3^, 10^4^, and 10^5^ copies/μL. The average relative detection accuracy was 83.43%, 87.69%, and 87.77%, respectively. These results validate the system’s good detection stability in cross-reaction conditions. Based on the results of these three experiments, the detection system demonstrates good reliability.

Additionally, the detection system was designed to detect target DNA only and lacked field detection capabilities. It lacks a fungal lysis chamber, making on-site detection of *Pyricularia grisea* challenging. Therefore, further research is needed to add lysis chambers and DNA extraction chambers to the microfluidic chip, with the aim of improving the on-site detection performance of the detection system [25]. The reaction chamber of this detection system is sealed under the action of flow-blocking valves, and the temperature of the heated reaction chamber can be precisely controlled within the range of 35 °C to 90 °C. Therefore, this microfluidic detection system can not only be applied to the rapid detection of *Pyricularia grisea* but also holds potential for extension to the isothermal amplification-based detection of other pathogens, demonstrating broad application prospects. The detection system can be integrated with a smartphone to develop a mini program for the automatic calculation of detection results. This application can automatically fit the detection data to a standard curve to determine the DNA concentration and automatically transmit the data to a smartphone for convenient viewing.

## Figures and Tables

**Figure 1 sensors-25-02511-f001:**
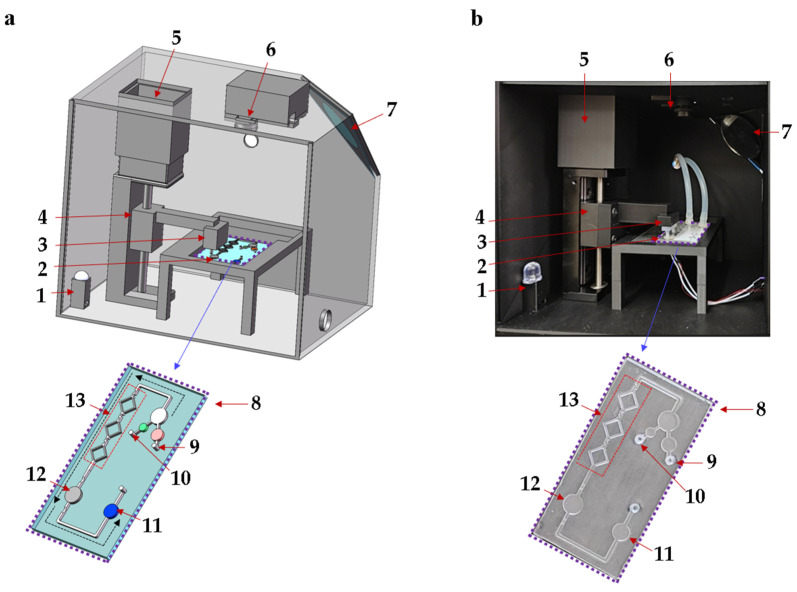
Schematic diagram of the microfluidic biosensing detection system for *Pyricularia grisea*. (**a**) Detection system structure. (**b**) Photograph of detection system. 1: Light bead, 2: Ceramic heater, 3: Control valve, 4: Vertical gearing, 5: Motor, 6: OpenMv camera, 7: Flat mirror, 8: Microfluidic chip, 9: DNA sample inlet, 10: LAMP reagent inlet, 11: Detection chamber, 12: Reaction chamber, 13: Micro-mixing channels.

**Figure 2 sensors-25-02511-f002:**
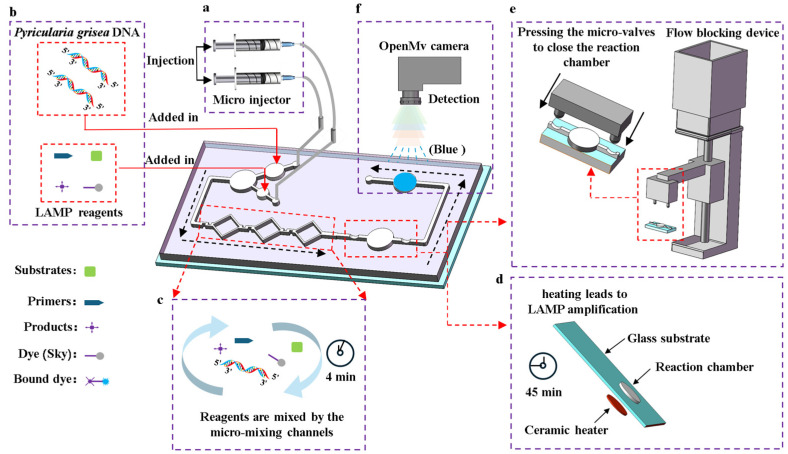
Detection process of the microfluidic detection system for *Pyricularia grisea*. (**a**) A micro injector drives the flowing of the reagents. (**b**) Loading reagents. (**c**) The reagents were mixed using the micro-mixing channels. (**d**) LAMP amplification is performed in the reaction chamber. (**e**) The flow blocking device seals the reaction chamber. (**f**) Image acquisition from the detection chamber by using the OpenMv camera.

**Figure 3 sensors-25-02511-f003:**
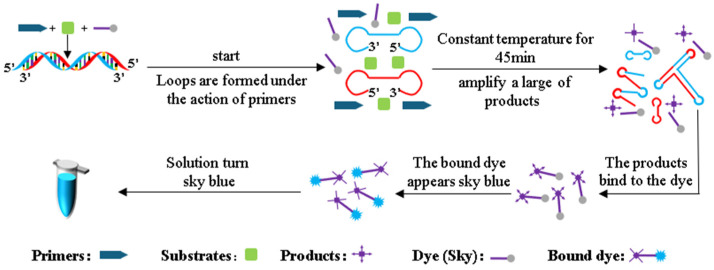
LAMP detection principle.

**Figure 4 sensors-25-02511-f004:**
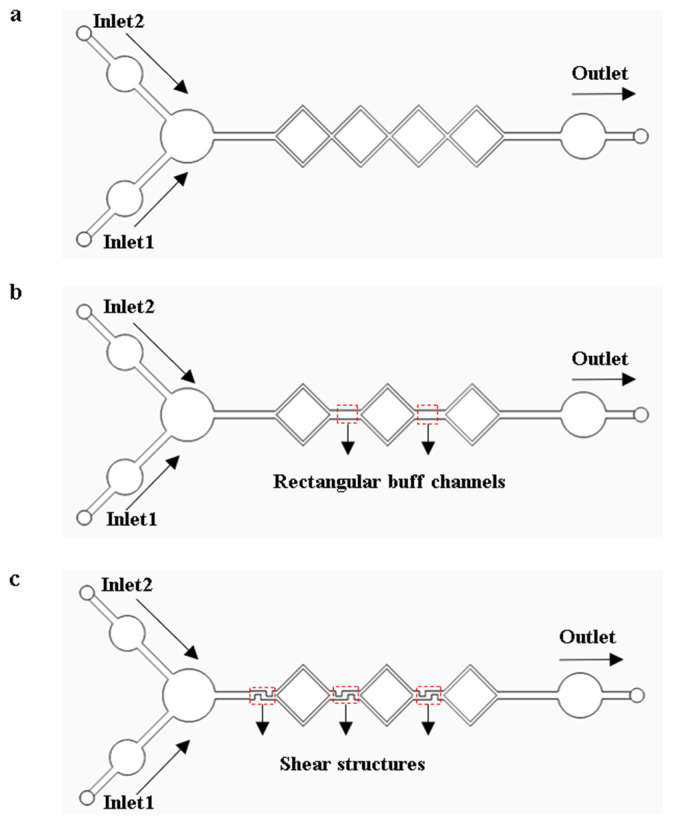
Micro-mixing channel models. (**a**) Square micro-mixing channel model. (**b**) Model with rectangular buffer channels. (**c**) Model with shear structures.

**Figure 5 sensors-25-02511-f005:**
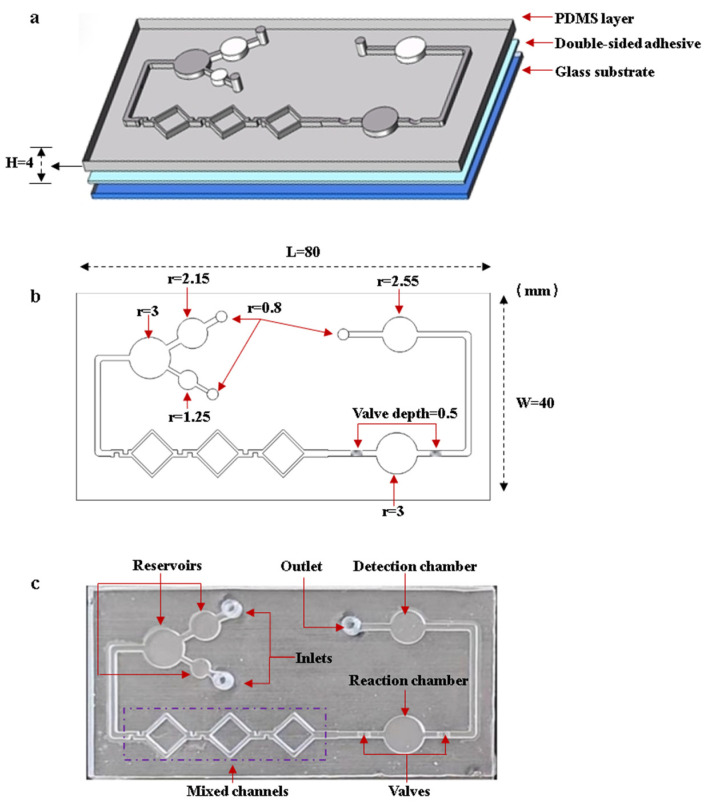
Designed structures and the fabricated microfluidic chip. (**a**) Structures of the microfluidic chip. (**b**) Dimensions of microfluidic chip. (**c**) Image of the microfluidic chip.

**Figure 6 sensors-25-02511-f006:**
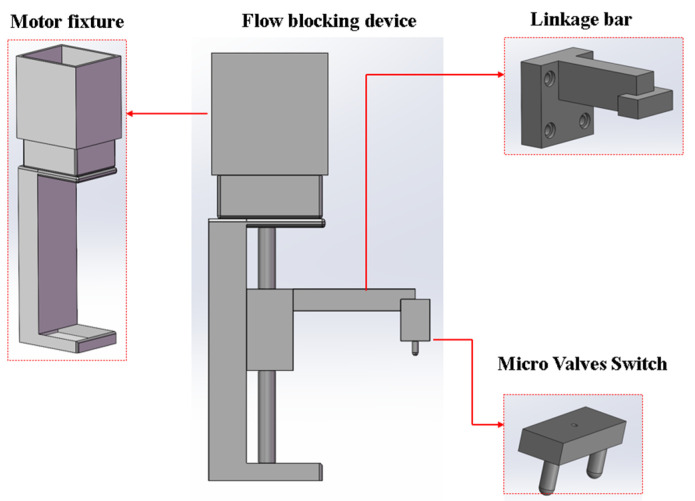
Structure models of the parts of the flow-blocking device.

**Figure 7 sensors-25-02511-f007:**
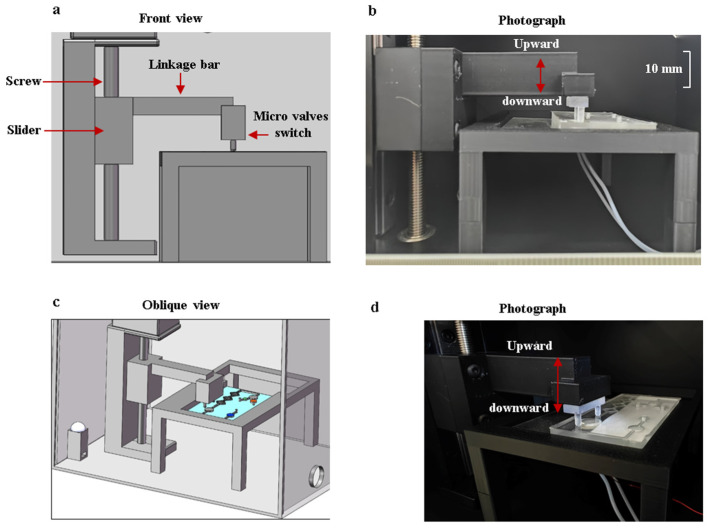
The collaborative operation of the flow-blocking device and the microfluidic chip. (**a**) Front view of the structural model. (**b**) Frontal photo of the flow-blocking device with a scale. (**c**) Oblique view of the structural model. (**d**) Oblique view photo of the flow-blocking device.

**Figure 8 sensors-25-02511-f008:**
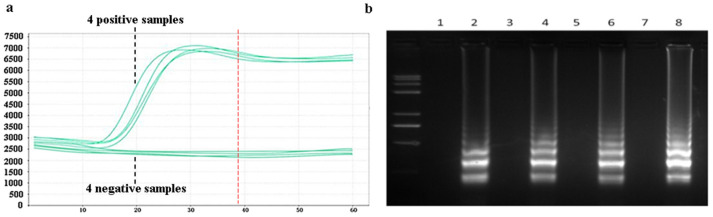
Specificity analysis using a PCR machine. (**a**) Amplification curves of negative and positive samples in the PCR machine. (**b**) Corresponding electrophoretic bands of the negative and positive samples, where odd-number bands are the four negative samples and even-number bands are the electrophoretic bands of the four positive samples.

**Figure 9 sensors-25-02511-f009:**
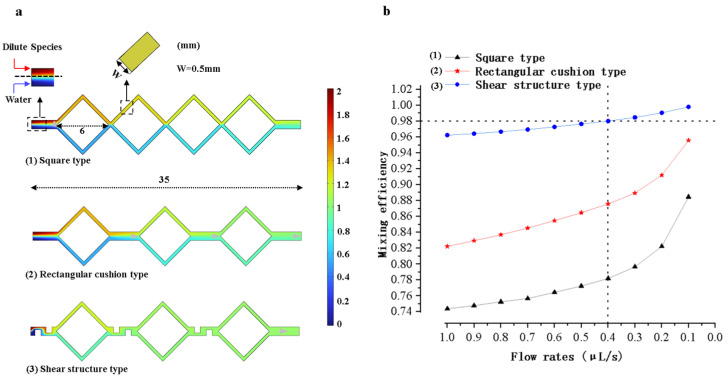
Mixing microchannel simulation results and analysis. (**a**) Graphs of the simulation results of microchannels with three different structures. (**b**) Mixing efficiency versus flow rates for three different structural models.

**Figure 10 sensors-25-02511-f010:**
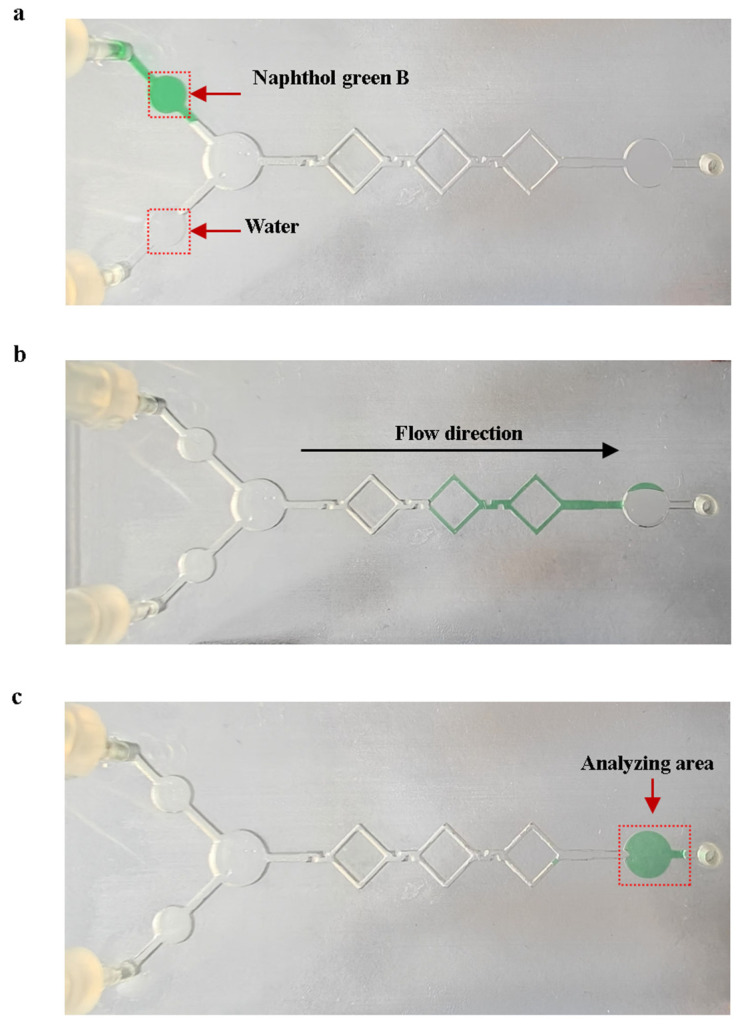
Mixing experiments using naphthol Green B with water. (**a**) State of the solution on the mixing chip at the start of mixing. (**b**) State of the solution in the micro-mixing channels during the mixing process. (**c**) State of the mixed chip at the end of mixing.

**Figure 11 sensors-25-02511-f011:**
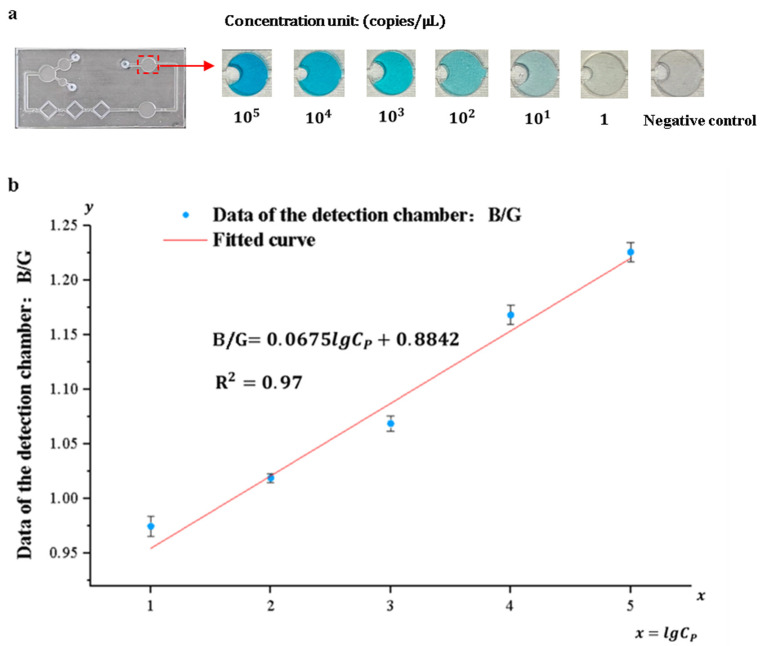
Detection of *Pyricularia grisea* using a microfluidic detection system. (**a**) Images of the detection chamber. (**b**) The fitted curve of B/G. (**c**) The fitted curve of B/R.

**Figure 12 sensors-25-02511-f012:**
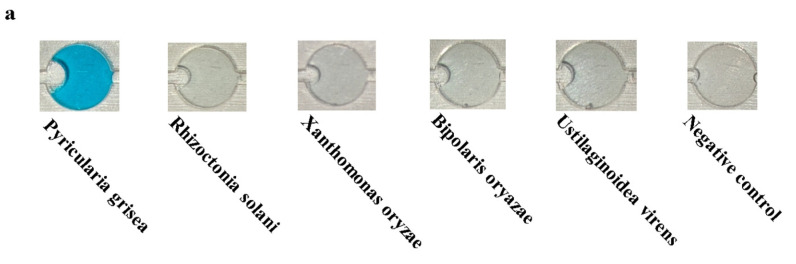
Specificity experiment results for the detection system. (**a**) Images of the chip detection chamber. (**b**) Detection system specificity assessment.

**Table 1 sensors-25-02511-t001:** LAMP primer sequence.

Primer Name	Sequence (5′ to 3′)	Target Gene
F3	CTGGGATGTGGTGACTTGC	STD
B3	AAGAACACCACCCGAGGT	STD
FTP	CGGCTCAAAGTTCCTCACGG ATAACCCGCCCTACATATCAGA	STD
BIP	CGGGACCTAGGCGGGTTGTAC GACACATGTCCAGCACTC	STD
LF	TCGCGTGCGAAACCTGC	STD
LB	ACAGTTAGGCCCGAGGACG	STD

**Table 2 sensors-25-02511-t002:** The detection results for samples containing only *Pyricularia grisea* DNA and samples after interference mixing.

*Pyricularia grisea* Only (copies/μL)	Detection Values (B/G)	Mixed DNA Solutions (copies/μL)	Detection Values (B/G)	Standard Deviation
10^5^	1.216	10^5^	1.253	1.85%
10^4^	1.152	10^4^	1.191	1.95%
10^3^	1.078	10^3^	1.112	1.70%

**Table 3 sensors-25-02511-t003:** Results of *Pyricularia grisea* DNA concentration detection in the mixed solution.

C_R_ (copies/μL)	T_1_	T_2_	T_3_	A_1_	A_2_	A_3_	$\bar{A}$
10^5^	87,635	119,792	117,543	87.64%	80.21%	82.45%	83.43%
10^4^	9327	8261	11,282	93.27%	82.61%	87.18%	87.69%
10^3^	1028	1166	827	97.20%	83.40%	82.70%	87.77%

## Data Availability

Data are contained within the article.

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
