# Peer review of "A LAMP Detection System Based on a Microfluidic Chip for Pyricularia grisea"

_sensors, 2025, doi:10.3390/s25082511_

Round 1

Reviewer 1 Report

Comments and Suggestions for Authors

I read the manuscript with interest. The manuscript titled "A LAMP Detection System Based on Microfluidic Chip for Pyricularia grisea" presents an innovative approach to the detection of Pyricularia grisea using loop-mediated isothermal amplification (LAMP) integrated within a microfluidic chip system. The authors have developed a robust, portable, and sensitive method suitable for rapid pathogen detection, contributing positively to disease management practices. However, there are several critical aspects that should be addressed to enhance the clarity, reproducibility, and comprehensiveness of the manuscript:

Comment 1: The COMSOL model code utilized for simulating the micro-mixing channels is essential for validation and reproducibility of results. It is strongly recommended to upload the COMSOL model files as supplementary materials to facilitate verification and replication by other researchers.

Comment 2: Several figures in the manuscript, notably Figures 5, 8, and 9, appear small, lack clear resolution, and importantly, do not include scale bars. Providing higher-resolution images along with clearly marked scale bars would greatly enhance the visual clarity and interpretability of these figures, enabling readers to precisely evaluate experimental results.

Comment 3: The manuscript currently lacks a comprehensive explanation and schematic diagrams describing the fundamental principles of the detection method in sufficient detail. It is highly advisable to include clear, detailed schematics and thorough descriptions of the underlying detection principles and their operational processes. This would significantly improve reader understanding, especially for readers unfamiliar with LAMP technology or microfluidic systems.

Comment 4: The paper should clearly describe the potential limitations and practical challenges of deploying the proposed microfluidic chip-based system in real-world or field conditions. This includes discussing potential variability in performance under different environmental conditions, sample preparation variations, and operational reliability in non-laboratory settings.

Comment 5: While specificity and anti-interference tests are presented, additional information on cross-reactivity and robustness tests under varied sample conditions (e.g., impurities, inhibitors present in real samples) would greatly enhance the manuscript’s thoroughness and practical applicability.

Reviewer 2 Report

Comments and Suggestions for Authors
  1. Please correct spelling.
  2. In Figure 1 description on the right is trimmed - please correct.
  3. According to the description, bottom of the microfluidic structures was made of double-adhesive tape, while the rest was a solid PDMS. These materials differ, so please explain how it influenced the flow and reactions. How have you implemented this material difference in your simulation model?
  4. Please explain some constructional details of the flow-blocking valves. They currently are described more by action not the structure.
  5. Some sentences in the chapter 3 are described as citation from manual: "add sample", "calculate efficiency" etc. Please unify the style.

Reviewer 3 Report

Comments and Suggestions for Authors

Comments attached.

Round 2

Reviewer 1 Report

Comments and Suggestions for Authors

Thank you for addressing the comments. I do not have further comments.